# Acute Stress Regulates Sex-Related Molecular Responses in the Human Jejunal Mucosa: Implications for Irritable Bowel Syndrome

**DOI:** 10.3390/cells12030423

**Published:** 2023-01-27

**Authors:** Bruno K. Rodiño-Janeiro, Marc Pigrau, Eloísa Salvo-Romero, Adoración Nieto, Elba Expósito, Ana M. González-Castro, Carmen Galán, Inés de Torres, Teodora Pribic, Laura Hernández, Beatriz Lobo, Marina Fortea, Milagros Gallart, Cristina Pardo-Camacho, Danila Guagnozzi, Javier Santos, Carmen Alonso-Cotoner

**Affiliations:** 1Department of Gastroenterology, Vall d’Hebron Hospital Universitari, Vall d’Hebron Barcelona Hospital Campus, Passeig Vall d’Hebron 119-129, 08035 Barcelona, Spain; 2Laboratory of Neuro-Immuno-Gastroenterology, Digestive System Research Unit, Vall d’Hebron Institut de Recerca (VHIR), Vall d’Hebron Hospital Universitari, Vall d’Hebron Barcelona Hospital Campus, Passeig Vall d’Hebron 119-129, 08035 Barcelona, Spain; 3Laboratory of Translational Mucosal Immunology, Vall d’Hebron Institut de Recerca (VHIR), Vall d’Hebron Hospital Universitari, Vall d’Hebron Barcelona Hospital Campus, Passeig Vall d’Hebron 119-129, 08035 Barcelona, Spain; 4Facultad de Medicina, Universitat Autònoma de Barcelona, 08193 Barcelona, Spain; 5Department of Pathology, Vall d’Hebron Hospital Universitari, Vall d’Hebron Barcelona Hospital Campus, Passeig Vall d’Hebron 119-129, 08035 Barcelona, Spain; 6Centro de Investigación Biomédica en Red de Enfermedades Hepáticas y Digestivas (CIBERHED), Instituto de Salud Carlos III, 28029 Madrid, Spain

**Keywords:** acute stress, sex, intestinal barrier, human, irritable bowel syndrome

## Abstract

Irritable bowel syndrome (IBS) is a prevalent gastrointestinal disorder linked to intestinal barrier dysfunction and life stress. We have previously reported that female sex per se determines an increased susceptibility to intestinal barrier dysfunction after cold pain stress (CPS). We aimed to identify sex-related molecular differences in response to CPS in healthy subjects to understand the origin of sex bias predominance in IBS. In 13 healthy males and 21 females, two consecutive jejunal biopsies were obtained using Watson’s capsule, at baseline, and ninety minutes after CPS. Total mucosal RNA and protein were isolated from jejunal biopsies. Expression of genes related to epithelial barrier (*CLDN1, CLDN2, OCLN, ZO-1, and ZO-3*), mast cell (MC) activation (*TPSAB1, SERPINA1*), and the glucocorticoid receptor (*NR3C1*) were analyzed using RT-qPCR. *NR3C1, ZO-1* and *OCLN* protein expression were evaluated through immunohistochemistry and western blot, and mucosal inflammation through MC, lymphocyte, and eosinophil numbering. Autonomic, hormonal, and psychological responses to CPS were monitored. We found an increase in jejunal MCs, a reduced *CLDN1* and *OCLN* expression, and an increased *CLDN2* and *SERPINA1* expression 90 min after CPS. We also found a significant decrease in *ZO-1, OCLN*, and *NR3C1* gene expression, and a decrease in OCLN protein expression only in females, when compared to males. CPS induced a significant increase in blood pressure, plasma cortisol and ACTH, and subjective stress perception in all participants. Specific and independent sex-related molecular responses in epithelial barrier regulation are unraveled by acute stress in the jejunum of healthy subjects and may partially explain female predominance in IBS.

## 1. Introduction

The influence of life stress in the development and severity of clinical conditions characterized by chronic pain and behavioral and psychological abnormalities is undisputed [1,2,3,4]. Intriguingly, many of these conditions show an unexplained female predominance that, until recently, has not been properly addressed. Irritable bowel syndrome (IBS) is one of such chronic conditions displaying distorted brain–gut communication characterized by abdominal pain and a change in bowel habits in frequent association with psychological alterations [5]. IBS is one of the most prevalent gastrointestinal disorders, affecting 4.1% of the general population with a female predominance particularly in Western countries (1.7 odds ratio) [6]. Gastrointestinal infections, diet, and life stress are common etiopathogenic factors in IBS [7,8,9] along with genetic predisposition and gut microbiota alterations [10,11]. In the last years, different observations in IBS patients, with predominant diarrhea (IBS-D), have found intestinal barrier dysfunction, immune activation, and low-grade mucosal inflammation in both the small and large intestine [12,13,14,15,16,17,18]. Specifically, some IBS-D patients display decreased expression of zonula occludens-1 (ZO-1) and increased expression of claudin-2 (CLDN2), reduced occludin (OCLN) phosphorylation, enhanced myosin light chain phosphorylation and dilatation of paracellular spaces, and condensation of cytoskeleton structures in the cell junctions between enterocytes at the jejunal mucosal [19,20]. These molecular and structural changes in epithelial tight junctions (TJs) in the small intestine have been linked to mast cell (MC) activation [14,19,20] and also to enhanced intestinal permeability [13,21].

There is also converging evidence that stress can impair intestinal barrier regulation [22] and induce MC activation [13]. Both acute and chronic stress have been shown to increase ion and water secretion, and enhance transcellular and paracellular permeability of small and large molecules involving MCs and neural mechanisms [22]. In human studies, acute physical or psychological stress increased ion and water secretion in healthy subjects and patients with food allergy [23,24,25] through parasympathetic pathways involving corticotropin releasing factor (CRF) and MC activation [13,21,25]. Stress also activates the hypothalamic–pituitary–adrenal (HPA) axis, leading to the release of glucocorticoids, which in turn downregulate CRF expression through the glucocorticoid nuclear receptor, subfamily 3, group C, member 1 (NR3C1), under a negative feedback mechanism [22]. Interestingly, NR3C1 has recently been shown to regulate TJ protein Claudin-1 promoter in an animal model [26], suggesting that NR3C1 could play a role in stress-induced intestinal barrier dysfunction in humans.

We have previously described that female sex per se determines an increased intestinal permeability in response to acute experimental stress in healthy subjects when compared to males, a mechanism that could contribute to female susceptibility to IBS [27]. However, the underlying molecular mechanisms have not yet been elucidated. Hence, in the present work, we aim to identify the molecular changes involved in the differential sex-determined changes in human intestinal permeability in response to acute stress in healthy individuals, as an initial step to understanding the origin of IBS’s female predominance.

## 2. Materials and Methods

### 2.1. Participants

Healthy subjects (HS) ranging 18–50 years old were prospectively recruited through public advertising. Prior to entering the study, potential candidates were asked to fill out the modified social readjustment scale of Holmes–Rahe to evaluate significant life events in the last year [28], the perceived stress scale (PSS) of Cohen [29] to assess levels of stress in the last month, and the Beck’s inventory for depression to assess depression levels in the last week [30], and only subjects with low baseline stress levels (Holmes–Rahe score < 150) and without signs of depression (Beck’s score < 9) were included. Medical history and a physical examination were carried out in all candidates. Exclusion criteria included acute gastroenteritis in the last two years, active smoking, abnormal menstrual cycle, pregnancy, contraceptive use, and any chronic health disorders. Food allergies were ruled out using a battery of skin prick tests (Leti S.A., Madrid, Spain) for 22 common foodstuffs and 12 inhalants, with histamine and saline as positive and negative controls, respectively. Subjects were not allowed to take salicylates, nonsteroidal anti-inflammatory drugs, anticholinergic drugs, or opioids 15 days prior to the biopsy. None of the participants were taking probiotics in the two months prior to inclusion.

The study protocol was approved by the Ethics Committee at Hospital Vall d’Hebron (PR (IR) 248/2012) and conducted according to the revised Declaration of Helsinki. All subjects provided their written informed consent.

### 2.2. Cold Pain Stress (CPS)

Acute experimental stress was induced by the cold-water pressor test [31]. Briefly, participants immersed the non-dominant hand in iced water (4 °C) for periods of 45 s followed by 15 s withdrawal intervals to prevent adaptation to pain for a total time of 15 min.

To assess whether the changes observed could be related to the intubation itself, a sham protocol was performed using the same procedures but without CPS. Intestinal biopsies, blood samples and processing of samples were performed identically in both groups.

### 2.3. Jejunal Biopsy

After an overnight fast, the Watson’s capsule was orally inserted under fluoroscopic control, to the proximal jejunum, 5 cm distal to the Treitz’s angle. Two consecutive mucosal biopsies were obtained: the first, at baseline, and the second 90 min after completing a 15 min-period of CPS. Tissue samples were immediately split into two similar pieces each with a sterile scalpel. One fragment was fixed in formalin and embedded in paraffin for histology and immunohistochemistry. The other fragment was placed in RNA later (Ambion, Invitrogen), kept at 4 °C for two hours and stored at −80 °C until processed for RNA isolation for RT-qPCR and protein isolation for western blot (WB).

### 2.4. Systemic Response to CPS

#### 2.4.1. Hand Pain Perception

The level of hand discomfort/pain was assessed using a visual analogue scale from 0 (no discomfort) to 10 (intolerable pain).

#### 2.4.2. Autonomic Response

Autonomic response was evaluated by measuring blood pressure and heart rate with an automated sphygmomanometer (Omron M4-I, Omron Healthcare Europe B.V., Hoofddorp, The Netherlands).

#### 2.4.3. Psychological Response

The level of acute stress experienced by participants was evaluated using the subjective stress rating scale (SSRS) [32].

#### 2.4.4. Hormonal Response

HPA-axis activation was assessed through plasma adrenocorticotropic hormone (ACTH) levels, determined using sandwich chemiluminescence immunoassay (LaisonXL, DiaSorin S.p.A., Saluggia, Italy), and cortisol levels, measured by chemiluminescent immunoassay (ADVIA Centaur Cortisol assay, Siemens Healthcare Diagnostics, Munich, Germany). Blood samples were collected in plastic tubes (BD Vacutainer^®^ Plus Plastic K_2_-EDTA Tubes, Franklin Lakes, NJ, USA), centrifuged, and aliquoted for hormonal determinations.

### 2.5. Intestinal Response to CPS

#### 2.5.1. Molecular Response

Mucosal molecular response to CPS was assessed by measuring gene expression levels of the following genes using qPCR (defined in Appendix A, described in detail in Supplementary Material and Methods): claudin 1 (*CLDN1*), claudin 2 (*CLDN2*), occludin (*OCLN*), zonula occludens 1 (*ZO-1*), and zonula occludens 3 (*ZO-3*) as markers of epithelial function; MC activation-related genes such as tryptase alpha/beta 1 (*TPSAB1*) and serpin family A member 1 (*SERPINA1*); and the glucocorticoid receptor nuclear receptor subfamily 3 group C member 1 (*NR3C1*) as a regulator of the stress response. To compare gene expression differences between groups, Ct values for each experimental group were determined, involving a target gene and a housekeeping gene (PPIA) that were subtracted to obtain the fold change 2 − ΔΔCt (ΔΔCt = ΔCt_Post-stress_ − ΔCt_pre-post_ = (Ct target gene_Post-stress_ − Ct PPIA_Post-stress_) − (Ct target gene_Pre-stress_ − Ct PPIA_Pre-stress_)). For the basal expression levels relative to reference, gene (PPIA) levels were calculated as 2 − ^ΔCt^ (ΔCt = (Ct target gene_Pre-stress_ − Ct PPIA_Pre-stress_)) and represented as log10(deltaCt).

In addition, ZO-1 and OCLN protein expression was analyzed using WB (described in detail in Appendix A). N3CR1 expression was assessed through immunohistochemistry (explained in detail in the Appendix A, anti-NR3C1 antibody described in Appendix A).

#### 2.5.2. Mucosal Inflammation and NR3C1 Expression

Jejunal low-grade inflammation was evaluated by numbering mucosal MC, eosinophils, and lymphocytes. Tissue sections fixed in formalin were stained with routine hematoxylin and eosin procedures and an expert pathologist blindly assessed epithelial morphology and eosinophilic infiltration. In addition, the number of intraepithelial lymphocytes and MCs was determined at ×400 magnification after immunohistochemical staining using anti-human CD3 and c-kit (CD117) antibodies (DAKO, Agilent, Santa Clara, CA, USA) (used antibodies described in Appendix A), respectively, as previously described [12].

### 2.6. Experimental Design and Procedures (Figure 1)

Physical examination, allergy testing, and baseline stress and depression assessments were performed the week prior to the jejunal biopsy. After an overnight fast, participants were orally intubated at 8:00 h. A baseline biopsy was collected after adequate placing of the Watson’s capsule 5 cm distal to the Treitz’s angle. The capsule was withdrawn and, as it only allows one biopsy per procedure, subjects were again intubated, submitted to the CPS protocol and 90 min after finishing CPS, a second biopsy from the same location was taken.

Blood samples (20 mL) were collected before oral intubation (t−95), after baseline biopsy (t−45), immediately before initiating CPS (t0), 5, 10, and 15 min after stress initiation (t5, t10, t15) and 30 (t45) and 90 (t105) minutes after stress cessation. Autonomic and psychological responses were measured before oral intubation (t−95), before (t−65), and after baseline biopsy (t−45), immediately before initiating CPS (t0), 5, 10, 15, and 20 min after stress initiation (t5, t10, t15, t20) and 30 (t45) and 90 (t105) minutes after stress cessation. Hand pain perception was assessed every 5 min during the stress period, and 5 and 30 min after stress cessation.

### 2.7. Statistical Analysis

Data from systolic blood pressure (SBP), diastolic blood pressure (DBP), heart rate (HR), subjective stress rating scale (SRSS), hand pain, ACTH, cortisol, and immune cells are expressed as median with first and third quartiles [Q1–Q3] and compared using a two-way repeated-measure ANOVA where sex was considered the between-subjects factor, and changes throughout perfusion time the within-subject factor (time). Differences between the different time points were analyzed with paired t-test corrected by Bonferoni as post hoc analysis.

Baseline expression levels compared with the reference gene (*PPIA*) are expressed as median with first and third quartiles (Q1–Q3) and the differences by sex were parametric and analyzed with an unpaired *t*-test. Fold change expressions after and before stress corrected by the reference gene (*PPIA*) are expressed as median with first and third quartiles (Q1–Q3). Fold change effects by stress were analyzed using one sample *t*-test (compared with the value of 1). Fold change effects by stress and sex were analyzed using Mann–Whitney U test for *ZO3* and tryptase (*TRYP*) (no parametric variables) or unpaired t-test for the rest of genes (parametric variables).

Data were scaled (“scale” command in R) for correlation plots and clustered by hierarchical clustering with the “hclust” command in R, and the method “complete”. Regression lines from the correlation dot plots of variables by pairs were calculated with the “geom_smooth” command in R, and with the “loess” method; the grey area represents the 95% confidence interval. *p* values < 0.05 were considered significant. In the figures, * represents *p* value < 0.05 and ** represents *p* value < 0.01. 

All data were analyzed using commercial software (SPSS 22.0, IBM, Armonk, NY, USA) and RStudio (RStudio v 1.1.419, PBC, Boston, MA, USA). All authors had access to the study data and had reviewed and approved the final manuscript.

## 3. Results

### 3.1. Demographics, Baseline Stress, and Depression Levels

Thirty-seven HS were initially recruited and submitted to the experimental protocol (Figure 1). Three participants were excluded from the study (two were unable to end the protocol, another presented significant intraepithelial lymphocytosis, Figure 2). Six participants were excluded from RNA expression analysis due to low RNA quality. Protein expression experiments were only performed on 11 subjects due to limited sample availability (Figure 2).

No demographic or psychological (stress and depression levels) differences between male (M) and female (F) groups were observed in the remaining 34 eligible subjects (13 males, 21 females, Table 1).

A sham protocol was performed on seven participants (three males and four females). Demographic data of the participants included in the sham protocol are shown in Appendix A.

### 3.2. Systemic Response to CPS

#### 3.2.1. Autonomic Response

Baseline biopsy significantly increased diastolic blood pressure (DBP) [F(2, 43) = 5.202; *p* = 0.009] with no differences between M and F but had no effect on systolic blood pressure (SBP) [F(2, 43) = 1.891; *p* = 0.163] and heart rate (HR) [F(2, 43) = 0.360; *p* = 0.700] (Appendix A).

CPS significantly increased SBP [F(3, 95) = 7.602; *p* < 0.001] and DBP [F(3, 95) = 11.533; *p* < 0.0001], but not HR [F(3, 95) = 2.165; *p* = 0.0973] (Figure 3A) with no differences between M and F groups (SBP: [F(1, 31) = 2.288; *p* = 0.14], DBP: [F(1, 31) = 4.089; *p* = 0.0519], HR: [F(1, 31) = 2. 83; *p* = 0.103]).

In the sham stress protocol, baseline biopsy significantly increased DBP [F(2, 10) = 7.209; *p*= 0.0115] and HR [F(2, 10) = 5.187; *p*= 0.0285], but did not affect SBP [F(2, 10) = 1.968; *p* = 0.190] (Appendix A). Sham stress decreased SBP [F(3, 15) = 4.777; *p* = 0.0157] and HR [F(3, 15) = 3.440; *p* = 0.0441] (Appendix A).

#### 3.2.2. Psychological Response

The levels of stress experienced by participants (SSRS) increased after baseline biopsy, although this increase did not reach statistical significance [F(2, 34) = 2.698; *p* = 0.0817] (Appendix A).

CPS was associated with a significant enhancement in SSRS [F(3, 86) = 12.123; *p* < 0.0001] that was similar between M and F groups [F(1, 28) = 0.416; *p* = 0.524] (Figure 3A).

In the sham stress protocol, baseline biopsy did not significantly increase SSRS [F(2, 10) = 2.032; *p* = 0.182] (Appendix A) and sham stress did not affect SRSS either [F(3, 15) = 0.947; *p* = 0.443] (Appendix A).

#### 3.2.3. Hand Pain Perception

CPS increased hand pain perception in both groups [F(3, 93) = 101.84; *p* < 0.0001], and this increase was higher in females when compared to males [F(1, 31) = 4.72; *p* = 0.038] (Figure 3B).

#### 3.2.4. Hormonal Response

There were no differences in ACTH or cortisol values at baseline between M and F groups (U = 109, *p* = 0.669 for ACTH and U = 91, *p* = 0.259 for cortisol). Baseline biopsy was associated with a significant decrease in plasma ACTH [F(1, 30) = 5.973; *p* = 0.0206] and cortisol levels [F(1, 30) = 10.430; *p* = 0.003].

CPS induced a similar activation of HPA-axis in both groups in ACTH [F(3, 90) = 5.968; *p* < 0.001] and cortisol [F(3, 96) = 7,344; *p* < 0.001], with no differences between M and F groups ([F(1, 30) = 0,481; *p* = 0.493] and [F(1, 32) = 2,53; *p* = 0.122], respectively) (Figure 4).

In the sham protocol, baseline biopsy was associated with a significant increase in plasma ACTH [F(1, 5) = 21.06; *p* = 0.006] and cortisol [F(1, 5) = 7.984; *p* = 0.037] that decreased during sham stress period (ACTH: [F(3,1 5) = 8.50; *p* = 0.002]; cortisol: [F(3,1 5) = 20.412; *p* < 0.001]) (Appendix A).

### 3.3. Intestinal Response to CPS

#### 3.3.1. Mucosal Gene Expression

(a)Gene expression levels in the jejunal mucosa at baseline

Baseline expression compared with the reference gene is shown in Appendix A. When compared between sexes, baseline expression levels of *NR3C1* were significantly higher in females compared to males (F: −1.23 (−1.40–−1.17); M: −1.49 (−1.55–−1.29), *p* = 0.006), as were *OCLN* (F: −0.97 (−1.09–−0.92); M: −1.17 (−1.33–−1.09), *p* = 0.009) and *ZO-1* expression levels (F: −1.39 (−1.59–−1.34); M: −1.67 (−1.76–−1.46), *p* < 0.034) (Appendix A). There was an upward trend in baseline expression levels of *TRYP* in males (F: −1.50 (−1.57–−1.37); M: −1.25 (−1.32–−1.17), *p* = 0.067). No significant differences were found in the basal expression levels of *CLDN1, CLDN2, ZO-3*, and *SERPINA1* between sexes. A clustering analysis of the correlations of the basal gene expression levels found that *NR3C1* clustered together with *OCLN* and *ZO-1* and, to a lesser degree, with *CLDN1* and *CLDN2* (Appendix A).

Baseline expression levels compared with the reference gene in the subjects submitted to the sham stress protocol is shown in Appendix A.

(b) Gene expression levels in the jejunal mucosa after CPS

Ninety minutes after CPS, we found a significant change in the expression levels of *CLDN1, CLDN2, OCLN*, and *SERPINA1* genes (Figure 5A). The expression of *CLDN1* and *OCLN* was significantly reduced (*CLDN1*: 0.81 (0.74–0.98), fold change, *p* = 0.008; *OCLN*: 0.87 (0.76–1), fold change, *p* = 0.016), while *CLDN2* and *SERPINA1* expression significantly increased (*CLDN2*: 1.33 (0.94–1.7), fold change, *p* = 0.002; *SERPINA1*: 1.24 (1.04–1.45), fold change, *p* < 0.001).

Interestingly, after CPS, a differential response in gene expression between male and female groups was observed, as *NR3C1* expression was significantly reduced only in women, when compared to men (F: 0.88 (0.76–0.96); M: 1.05 (0.99–1.15), fold change, *p* = 0.007) (Figure 5B). Similarly, when we analyzed TJ gene expression, we found a significant decrease in *OCLN* gene expression, and this decrease was enhanced in women (F: 0.82 0.76–0.88); M: 0.99 (0.83–1.17), fold change, *p* = 0.022). In addition, we observed a differential regulation of *ZO-1* gene expression, with a decrease in women and a slight increase in men 90 min after CPS (F: 0.83 (0.73–0.98); M: 1.04 (0.96–1.12), fold change, *p* = 0.046) (Figure 5B).

A clustering analysis of the correlations of gene expression changes after CPS showed that *NR3C1* clustered together with *ZO-1, OCLN*, and *SERPINA1* (Figure 5C–G).

In the sham stress protocol, a significant reduction in the expression of *CLDN1* (*CLDN1*: 0.79 (0.77–0.82), fold change, *p* = 0.005) and an increase in the expression of *SEPINA1* (*SERPINA1*: 1.19 (1.12–1.41), fold change, *p* = 0.021) was observed (Appendix A)**.**

#### 3.3.2. Mucosal Inflammation and NR3C1 Protein Expression

At baseline, eosinophil (Eo) counts were higher in male when compared to female (Eo M: 3.2 (1.5–6.05) cells/hpf; Eo F: 1.30 (0.7–3.5) cells/hpf, *p* = 0.040) with no differences in MCs or CD3+ counts. 

Ninety minutes after finishing CPS we found an increase in MCs counts [F(1, 31) = 6,79; *p* = 0.014], with no differences between male and females [F (1,32) = 0.86; *p* = 0.362]. In contrast, Eo counts [F(1, 32) = 0,127; *p* = 0.724] and CD3+ counts did not change [F(1, 32) = 1,19 *p* = 0.282] after CPS (Figure 6A–C).

NR3C1 mean intensity by IHC in jejunal mucosal samples was similar at baseline between male and female groups (M 119.1 ± 2.3; F: 121.7 ± 1.535 mean intensity AU, *p* = 0.357) and we did not find significant modification after CPS (M: 122.1 ± 3.303, F: 121.4 ± 2.167) (Appendix A).

No significant changes in MCs, Eos, and CD3+ counts were observed in the sham stress protocol (Appendix A).

#### 3.3.3. TJ Protein Expression

After CPS, we found an increase in OCLN protein expression only in the male group, while there was a decrease in OCLN expression in the female group (M: 1.266 (1.072–1.799); F: 0.917 (0.738–1.036), fold change, *p* = 0.052) after CPS (Figure 7). ZO-1 protein expression was not affected by CPS (M: 0.873 (0.543–0.979); F: 1.019 (0.848–1.247), *p* = 0.177).

## 4. Discussion

The present work shows that sex determines the expression of TJs and stress-related genes/proteins in the human jejunal mucosa in response to acute stress. Experimental acute stress induced a differential regulation of gene expression levels between sexes in the human jejunal mucosa and increased the number of MCs, mainly in males. Moreover, the expression of TJ-related genes in males seems to be more resistant to stress-induced changes compared to their female counterparts. Finally, although not novel, we confirm that CPS induces a strong autonomic, hormonal, and psychological stress response, as previously described [27,33].

CPS is a well-known acute experimental stress model in humans [31] that induces both physical and psychological responses. Our previous studies have shown that CPS as well as the stress mediator CRF affect the intestinal barrier function in healthy subjects (HS) [13,27,33]. However, the molecular basis of these changes has not yet been completely elucidated.

There is no doubt that biological sex plays a role in the prevalence and severity of a number of important stress-related gastrointestinal and immune-related diseases such as IBS [2,34,35]. However, the underlying molecular mechanisms contributing to sex differences remain poorly understood. Our results show that acute stress downregulates gene expression of *CLDN1* and *OCLN*, while upregulating the expression of *CLDN2*. Interestingly, *ZO-1* and *OCLN* expression after CPS was different between male and female groups, being downregulated only in females. Similar findings have been described in different animal models of stress showing that acute stress is associated with a decreased expression of Cldn1, Ocln, and Nr3c1 and an increased expression of Cldn2 along with intestinal permeability and visceromotor changes in the colon [26,36,37,38,39]. In line with these results, we also found a higher stability of TJ gene expression in males in response to CPS, similarly to the strengthening of TJs function after pilocarpine stimulation reported in salivary glands of male mice when compared to females [40]. Of note, very few studies have focused on the small intestine. One of such studies showed that the combination of water avoidance and restraint stress downregulated mRNA and protein levels of ZO-1 in the duodenal mucosa of male rats [41]. While permeability studies in rodents have been performed only in male, a porcine model of early weaning stress showed a decrease in the expression of OCLN, CLDN1, and ZO-1 in the jejunal mucosa in both sexes in association with impaired intestinal barrier function [42]. ZO-1 plays an essential role in the control of paracellular permeability of large molecules through the leak pathway, as shown in MDCK monolayers after its knockdown through zonula adheren and zonula occluden belt regulation [43]. Our results show that a coordinated genetic regulation of TJs genes may also be involved, as suggested by the strong correlation between the expression of *ZO-1* and *NR3C1* in the human jejunum. Conversely, the role of Ocln in the paracellular integrity is controverted, as Ocln seems not essential for the regulation of intestinal barrier function and epithelial cell polarization in OCLN-deficient mice [44,45], while other studies indicate that OCLN can regulate the paracellular pore pathway in polarized Caco-2 cells [46], along with the passage of monovalent cations and uncharged solutes through the leak pathway in Caco-2 cells [47]. These actions of OCLN involve its ability to modulate strand architecture and CLDN channels and may support its regulatory role of barrier function under stressed conditions. Last, but not least, CLDN2 is a pore-forming molecule that facilitates paracellular flux of water [48], and Na+ and small uncharged molecules [49], whereas CLDN1 seems to play the opposite role [50]. Altogether, our findings suggest that females may be more susceptible to develop stress-related molecular alterations in the intestinal barrier leading to enhanced intestinal permeability in response to acute stress. This is in line with our previous results [13,27,33], where we showed an increased albumin secretion to the intestinal lumen in females as a marker of altered intestinal permeability. Albumin is a macromolecule normally transcellularly transported from the intestinal lumen to the blood. However, a bidirectional paracellular flux between the intestinal lumen and the lamina propria may occur through the leak pathway after immune activation [47,51]. Our present results may be an explanation for the higher increase in albumin secretion that we found in women when compared to males in response to CPS.

A novel finding of our study is the ability of acute stress to downregulate the expression of *NR3C1* in the human jejunum, only in females. Prior studies in male rats suggest a central role of glucocorticoids as mediators of the stress-induced enhancement of intestinal permeability as both adrenalectomy and the glucocorticoid receptor inhibitor RU-486 prevented the increase in permeability [52]. In addition, pretreatment with CRF increases the expression of CLDN2 in HT-29, T84, and MDCK cells [53]. HPA-axis response also seems essential for understanding sex-related differences in psychiatric disorders such as post-traumatic stress disorder and depression, but little is known about the role of the intestinal barrier in these disorders [54]. More studies are needed to ascertain the specific consequences of sex-dependent differential glucocorticoid receptor expression in the intestinal barrier function, their nuclear targets, and its clinical implication. Intriguingly, we also found that epithelial barrier genes correlate with *NR3C1* both at baseline, and after CPS. This is interesting, as the expression of *NR3C1*-correlated genes (i.e., *OCLN* and *ZO-1*) was different between sexes. Chronic stress limits the binding of NR3C1 to CLDN1 and OCLN promoters in Caco-2 BBE cortisol-treated cells, but binding of HES1 is not affected, leading to a decreased expression of Cldn1 in rat colon [26]. It would be interesting to explore whether acute stress activates a common nuclear mechanism leading to a coordinated change of *ZO-1*, *OCLN*, and *NR3C1* gene expression in the human jejunal mucosa and in both sexes.

Additionally, to the intestinal molecular response to acute stress, we also evaluated mucosal microinflammation and MC response, and found that stress increased MC counts and *SERPINA1* expression, but not *TRYP* mRNA expression, with no differences between sexes. Stress has been shown to increase MC degranulation and activation in the rat intestinal mucosa [55,56,57,58,59] as well as in humans [13,21,25], and several reports indicate that MCs play a key role in the regulation of intestinal barrier in animals and humans [21,56,58,59,60,61,62]. Among the mechanisms by which MCs can induce intestinal dysfunction, proteases have been shown to increase intestinal permeability through PAR2-mediated TJ regulation [59,63,64]. In fact, IBS patients showed higher protease activity in feces that has been related to impaired intestinal barrier function [65,66], particularly *TRYP* and trypsin-3 [63,67] signaling through epithelial *PAR2* cleavage and MC activation. Signaling through PAR2 leads to changes in localization and expression of TJ regulatory proteins [68]. *SERPINA1* encodes alpha-1-antitrypsin, one of the main human antiproteases that inhibits trypsin, tryptase, and neutrophil elastase, and it is produced by gut epithelial cells [69], specifically by enterocytes [70]. Interestingly, corticosteroid binding globulin (encoded by *SERPINA6*), a major high-affinity transporter for cortisol in the blood, is inactivated via proteolytic cleavage by neutrophil elastase, which, as mentioned before, is in turn inhibited by alpha-1-antitrypsin [71]. *SERPINA6* and *SERPINA1* are closely located in chromosome 14 and both were related in a meta-analysis of genome-wide association studies with morning plasma cortisol levels [72]. The lack of correlation between *SERPINA1* and *TRYP* gene expression levels and MCs count in our study raises the question on the meaning of the anti-protease gene upregulation in the control of barrier function. In this sense, *SERPINA1* gene expression could be increased to counter-regulate other proteases related with the dysfunctional intestinal barrier function, or alternatively, MCs from both sexes may react differently to stress, as has been shown in distal ileum of mice submitted to restraint stress [73]. Our results suggest a differential expression profile of MCs between sexes in response to stress. Sexually dimorphic responses have been previously reported in bone marrow MCs from mice submitted to restraint stress, with greater intestinal permeability and serum histamine responses to restraint stress in female mice [73]. In this study, female MCs showed a markedly increased capacity for synthesis and storage of granule-associated immune mediators, indicative of a sex-related differential transcriptome in MCs resulting from stress exposure [73]. However, more studies are needed to understand the role of sexual dimorphism in MC-mediated intestinal function in humans. 

We acknowledge that our work has several limitations mainly derived from the complexity of the methodology that limited the recruitment of participants, and adaptation of the time point for the second biopsy to technical feasibility that may explain the inability to explore genetic expression dynamics in more detail and lack of significant changes in protein expression, although the small sample size for protein expression experiments could also contribute to this point. Moreover, we cannot exclude that the effects on mucosal gene expression represent the combined effect of intubation, baseline biopsy, and CPS, as baseline biopsy was a stressful experience for participants by itself and we found changes in molecular expression of some genes in the sham protocol. Unfortunately, we could not prevent this response due to the complex experimental design. However, our study also has important strengths; notably, the performance of these in vivo experiments using a sophisticated and invasive procedure in human subjects that provides valuable information on the effect of acute stress in the intestinal mucosa in humans, although we acknowledge that this could be a limitation for reproducibility.

In conclusion, acute experimental stress uncovers a differential sex-related molecular response in the intestinal mucosa that may explain differences in stress-induced enhanced permeability between sexes. We speculate that these molecular alterations induced by acute stress could help to understand female susceptibility to develop IBS and possibly other gastrointestinal and systemic disorders related to life stress and brain–gut dysfunction.

## Figures and Tables

**Figure 1 cells-12-00423-f001:**
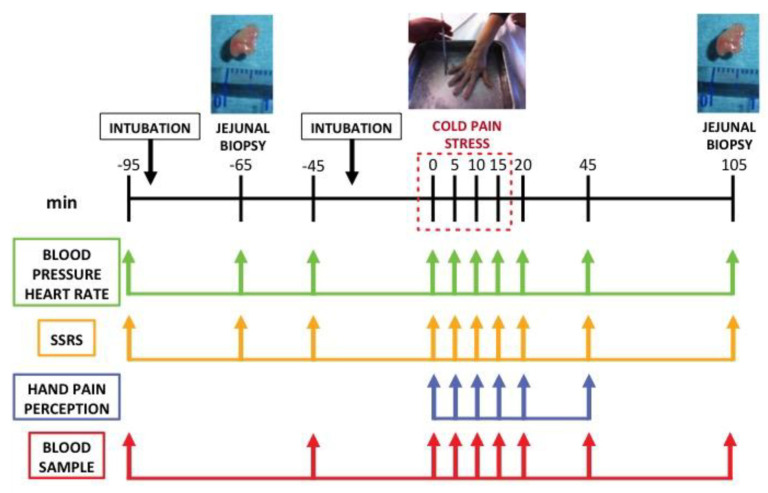
Experimental design and procedures. SSRS: subjective stress rating scale; Min: minutes.

**Figure 2 cells-12-00423-f002:**
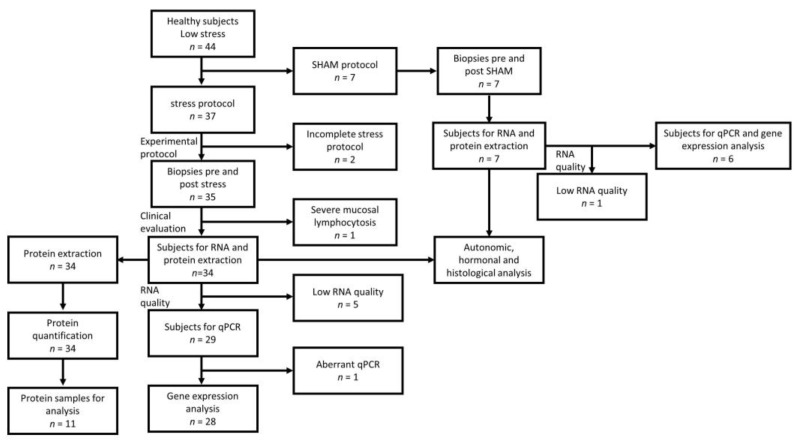
Flow chart of subjects’ inclusion and analysis.

**Figure 3 cells-12-00423-f003:**
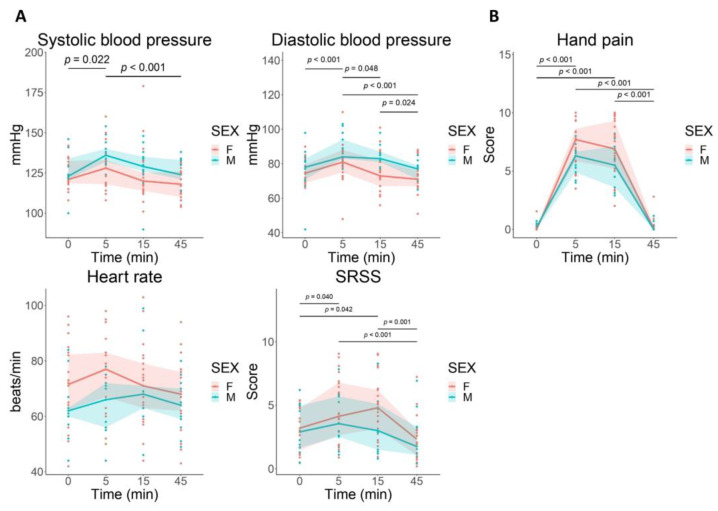
Systemic response to cold pain stress (CPS). The response to stress (change from baseline) was significantly different between males (M: blue circles) and females (F: pink circles). (**A**) A two-way ANOVA revealed significant differences for time and for sex group both in systolic and diastolic blood pressure. A two-way ANOVA revealed significant differences for time but not for sex in heart rate. CPS increased subjective stress rating scale (SSRS), with no differences between M and F groups. (**B**) A two-way ANOVA revealed significant differences for time and for sex group both in hand pain perception. Lines represent the median for each time point and the area represents the 25 and 75 percentiles. F: females; M: males; min: minutes; SSRS: subjective stress rating scale.

**Figure 4 cells-12-00423-f004:**
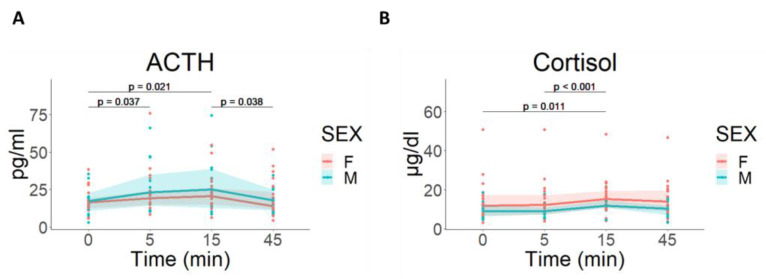
Hormonal response to CPS. (**A**) ACTH: A two-way ANOVA revealed significant differences for time, but not for sex groups. (**B**) Cortisol: A two-way ANOVA revealed significant differences for time, but not for sex groups. Males and females are represented with blue circles and pink circles, respectively. Lines represent the median for each time point and the area represents the 25 and 75 percentiles. ACTH: adrenocorticotropic hormone; CPS: cold pain stress; F: females; M: males; min: minutes.

**Figure 5 cells-12-00423-f005:**
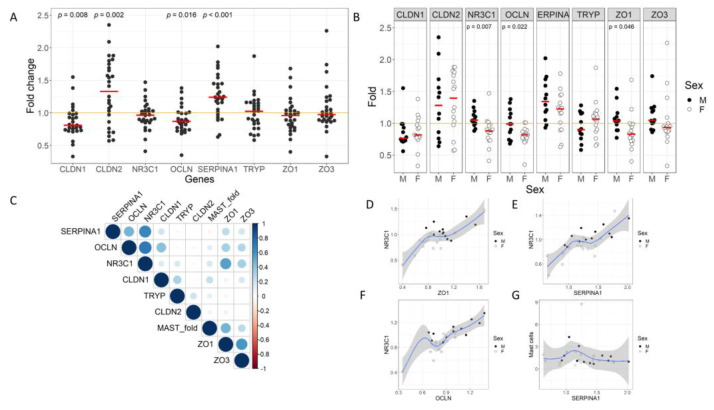
Gene expression levels of *CLDN1, CLDN2, NR3C1, OCLN, SERPINA1, ZO-1, ZO-3*, and *TRYP* in jejunal mucosa after CPS. (**A**) Fold change of gene expression levels induced by stress in jejunal mucosa. The red line indicates the median. (**B**) Fold change of gene expression levels induced by stress in jejunal mucosa by sex. The red line indicates the median. (**C**) Correlation graph with hierarchical clustering of the fold change in the gene expression levels and the number of MCs induced by stress. The dots size and color of the dots represent the degree of correlation (Spearman’s rho). (**C**–**G**) represent the correlation distribution of the fold change expression levels induced by stress among *NR3C1, ZO-1, SERPINA1*, and MC cell numbers. The blue line indicates the Loess regression and the gray area the confidence interval of 95%. Males and females are represented with back circles and white circles, respectively. F: females; M: males; *CLDN1:* claudin 1; *CLDN2*: claudin 2; *OCLN*: occludin; *ZO-1*: zonula occludens 1; *ZO-3*: zonula occludens 3; *TRYP*: tryptase; *SERPINA1*: serpin family A member 1; *NR3C1*: glucocorticoid receptor nuclear receptor subfamily 3 group C member 1.

**Figure 6 cells-12-00423-f006:**
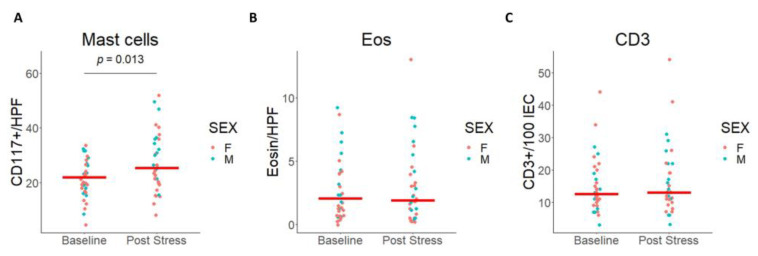
Mucosal inflammation in jejunal mucosa after CPS. (**A**) Mast cell counts before (Baseline) and after stress (Post Stress) expressed per HPF (400×). A two-way ANOVA revealed differences by stress, but not for sex. The red line indicates the median. Indicates *p* value = 0.013. (**B**) Eosinophil counts before (Baseline) and after stress (Post Stress) expressed per HPF. (**C**) Intraepithelial lymphocyte counts before (Baseline) and after stress (Post Stress) expressed per 100 intestinal epithelial cells. CD117: cluster of differentiation 117; HPF: high-power field; Eosin: eosinophils; CD3: cluster of differentiation 3; IEC: intestinal epithelial cells; F: females; M: males.

**Figure 7 cells-12-00423-f007:**
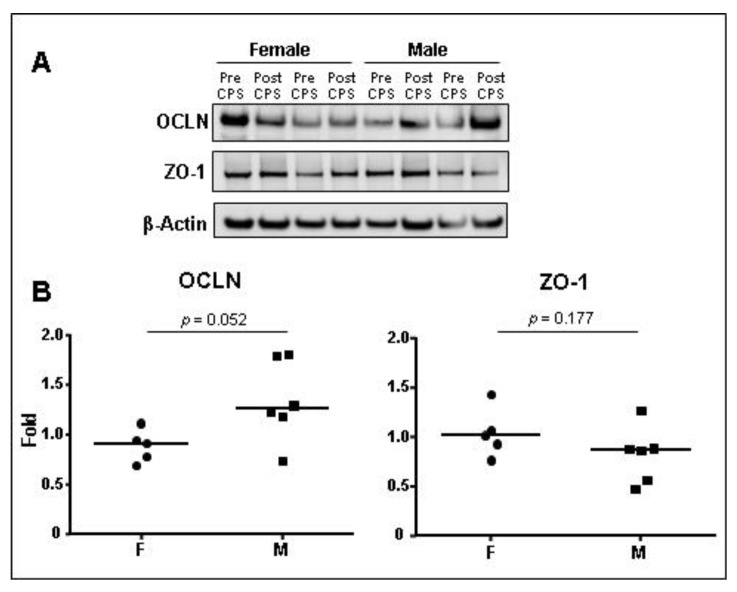
Protein expression levels OCLN and ZO-1 in jejunal mucosa after CPS. (**A**) Representative Western blot image showing expression of OCLN and ZO-1 in the jejunal mucosa of two representative females and two representative males before and after cold pain stress (CPS). Normalization was performed using β-actin as loading control. (**B**) Bands were quantified, and results are expressed as fold–change, each dot represents the fold change of each individual (five females and six males) before (Pre) and after CPS (Post). Intergroup comparisons were performed using the Mann–Whitney U test. *p* values are indicated. F: females; M: males; OCLN: occludin, ZO-1: zonula occludens 1.

**Table 1 cells-12-00423-t001:** Demographic characteristics of participants submitted to CPS protocol.

	Sex		
Variable	M (Q1–Q3)	F (Q1–Q3)	*p* Value
Age (years)	22.9 (22.1–29.2)	22.6 (21.7–25.2)	0.462
Holmes–Rahe scale (score)	76.0 (39.5–98.5)	89.0 (63.8–131.3)	0.420
Cohen’s scale (score)	14.0 (11.0–23.5)	19.0 (12.3–24.0)	0.529
Beck’s inventory (score)	0.0 (0.0–0.5)	0.5 (0.0–2.8)	0.737

Holmes–Rahe scale (to assess the level of stress over the last year); reference values: low stress <150), moderate 150–299, high ≥300. Cohen’s scale (to assess the level of stress over the last month); reference values: no perceived stress <19, mild perceived stress 19–28, moderate perceived stress 29–38, severe perceived stress >38. Beck’s depression inventory, to evaluate the level of depression during the last week; reference values: no depression <10, mild 10–18, moderate 19–29, severe depression 30–63. Data are expressed as median with first and third quartiles Q1–Q3). A Mann–Whitney U test was used for comparisons of continuous variables between groups.

## Data Availability

All the data, analytic methods, and study materials are present in the article and the Appendix A.

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
