# Peer review of "Acute Stress Regulates Sex-Related Molecular Responses in the Human Jejunal Mucosa: Implications for Irritable Bowel Syndrome"

_cells, 2023, doi:10.3390/cells12030423_

Round 1

Reviewer 1 Report

The authors already experts in the field of IBS one of the most common gastrointestinal disorders, which affects 4% of the general population worldwide, address a relevant gender issue as they demonstrate the female predominance.

Specifically they were able to demonstrate that CPS (cold pain stress) induced a significant decrease in ZO-1, OCLN, and NR3C1 gene expression, and a decrease in OCLN protein expression only in females, when compared to males. In particular, the responses to hormonal stress were evaluated with dosages of ACTH and cortisol.

Specifically, the authors could have studied the vagal remarriage by means of electromyography in order not to leave unresolved physiological tests.

But overall the premises have been well amply demonstrated with immunohistochemical and molecular methods with blotting

Reviewer 2 Report

Dear authors

I read with great interest you pilot study regarding  irritable bowel syndrome and your  aim to identify sex-related molecular differences in response to Cold Pain Stress in healthy subjects. The study is limited  due to the amount of recruited patients, but nevertheless, the paper is very well written and presented with proper English and Methodology and I endorse a publication, since it is going to contribute to elucidation of  the largely unknown field of functional bowel diseases and inspire further large-scale research to this direction

Reviewer 3 Report

 I read with great interest the manuscript entitled “Acute stress regulates sex-related molecular responses in the human jejunal mucosa. Implications for Irritable Bowel Syndrome ». The aim of this paper was to determine the molecular changes involved in the differential sex-determined changes in human intestinal permeability in response to acute stress in healthy individuals. The authors confirmed the previous results that CPS induces a strong autonomic, hormonal and psychological stress response. They also demonstrated that CPS induces a significant decrease in ZO-1 and OCLN and NR3C1 gene expression and a decrease in OCLN expression in females.

I only have few comments:

-        The authors should mentioned in the limitation that probiotics were authorized during the study. Probiotics can have an effect on microbiota but also on intestinal permeability. The fact that the first intubation could have an impact on the 1st biopsy while it is a stress for the volunteer.

-        The procedure to take jejunal biopsies with the Watson’s capsule should be more detailed. If I understand well it is done without anesthesia and without endoscopy? Why was the second intubation done before the cold pain stress? Because the volunteer has to keep the probe during the whole procedure, it is not very convenient! Also, the first intubation is shorter as the second?

-        I would like to have more explanation about the different questionnaires : homes, cohen and beck score. Number of questions, aim, higher score…

-        All the acronyms should be explained in the figures. For examples, SRSS in figure 2, ACTH in figure 3.

Otherwise, I do not have further comments and I congratulate the authors for their work.

Reviewer 4 Report

This is an intriguing study, in which the investigators should be congratulated for their success in getting so many healthy people through the invasive, arduous protocol. There are many problems with the data display and expression, in addition to several other issues. It is uncontrolled and I am surprised that this was not included in one of the limitations.

1.       Introduction: There are a few issues that are overstated. First, the female preponderance of IBS is not universal. Studies in Bangladesh and Pakistan have a preponderance of men. While the reason for this discrepancy is not understood (several theories), the authors’ statement of ‘worldwide’ should be toned down. Secondly, not all patients with IBS-D have low-grade inflammation, increased mast cell density  and impaired intestinal permeability. The absolute way this is presented has to be tempered to fit with the published data. Thirdly, not all studies have convincingly demonstrated stress induces increased intestinal permeability. While this might relate to the stress itself or the imprecision of measuring permeability and the authors are probably correct, this is only an assumption. Again, the unequivocal way the data are presented may be overstating the case.

2.       A key to the study is the intubation with a Watson capsule. This in itself is a stressful experience, but I did not see that addressed or discussed (apologies if I missed it). the fact that protein expression changed only after 90 minutes of the CPS was surprising and may relate to the initial stress followed by another stress. Did the authors assess the stress associated with the intubation? This highlights the limitation of the attribution of changes to the CPS when there is no placebo stress applied.

3.       The authors claimed that the biopsies were taken from the same place, but how did they know that? The statement is definite, but I think it should be qualified – with screening? Or just assumption?

4.       I do not understand that jejunal ‘low-grade’ inflammation was being examined. Surely, inflammation is being assessed and the emotive term ‘low grade’ is not needed? Also, please confirm (and state) that degranulated mast cells are counted with the methodology used.

5.       Fig 2: The x-axis is not labelled. It is likely that some subjects did not have what the authors have stated is a stress response. What did the authors define as a stress response and what proportion di not have this? Were those subjects examined in a post-hoc way for their response in tissue markers?

6.       Effect of CSP on mucosal gene expression is again not correct but represents the effect of intubation, biopsy and CPS. The results are awkwardly expressed. It would be better to sy all results are shown as fold changes and presumably they are median and IQR? The statistical comparisons made are not described as all possibilities are shown in the methods.

7.       Figure 5 states that it is about the mast cell density, but there are also eosinophil and lymphocyte densities. The legend needs to be expanded and which of the graphs is which stated so that the reader does not need to dig that information out of the text. Please state the actual p-value in the legend. In the text, the effect of stress on mast cell density in women is not stated.

8.       Supplementary Figure 1 shows the drop-off in numbers for gene expression, but does not state why protein expression is now only 11 subjects in Fig 6. The lack of statistical information in Fig 6 is notable. This part is unconvincing since there is not much time for the protein expression to change after the main stress (see above). The subject flow needs to be part of the main text.

9.       Table 1 needs explanation of the scores are since the ‘Homes score is not called that in the methods and I could not find what the Cohen score was. It should be in the Table legend.

10.   The supplementary methods are unsatisfactory as they do not tell me what the variation (CV) was between replicates and the housekeeper gene was’ PPIA’, which was not described (have to go to Table s1 for that) and the effect of stress on it not described. The RNA data are only as good as the housekeeper gene, so this need tidying/explanation.

11.   Figure s3> What is that supposed to show? There needs explanation as there is no obvious differences. I think these need pointing out.

Round 2

Reviewer 4 Report

Thank you for the modifications that are all appropriate and improve the manuscript